# International Perspectives of Prehospital and Hospital Trauma Services: A Literature Review

Rayan Jafnan Alharbi [1,2,*], Virginia Lewis [3] and Charne Miller [1,4]

1  School of Nursing & Midwifery, La Trobe University, Melbourne 3084, Australia; charne.miller@unimelb.edu.au
2  Department of Emergency Medical Service, Jazan University, Jazan 45142, Saudi Arabia
3  Australian Institute for Primary Care & Ageing, La Trobe University, Melbourne 3084, Australia; v.lewis@latrobe.edu.au
4  Department of Nursing, Faculty of Medicine, Dentistry & Health Sciences, The University of Melbourne, Melbourne 3010, Australia
*  Correspondence: ralharbi@windowslive.com

**Abstract:** Background: Evidence suggests that reductions in the incidence in trauma observed in some countries are related to interventions including legislation around road and vehicle safety measures, public behaviour change campaigns, and changes in trauma response systems. This study aims to briefly review recent refereed and grey literature about prehospital and hospital trauma care services in different regions around the world and describe similarities and differences in identified systems to demonstrate the diversity of characteristics present. Methods: Articles published between 2000 and 2020 were retrieved from MEDLINE and EMBASE. Since detailed comparable information was lacking in the published literature, prehospital emergency service providers' annual performance reports from selected example countries or regions were reviewed to obtain additional information about the performance of prehospital care. Results: The review retained 34 studies from refereed literature related to trauma systems in different regions. In the U.S. and Canada, the trauma care facilities consisted of five different levels of trauma centres ranging from Level I to Level IV and Level I to Level V, respectively. Hospital care and organisation in Japan is different from the U.S. model, with no dedicated trauma centres; however, patients with severe injury are transported to university hospitals' emergency departments. Other similarities and differences in regional examples were observed. Conclusions: The refereed literature was dominated by research from developed countries such as Australia, Canada, and the U.S., which all have organised trauma systems. Many European countries have implemented trauma systems between the 1990s and 2000s; however, some countries, such as France and Greece, are still forming an integrated system. This review aims to encourage countries with immature trauma systems to consider the similarities and differences in approaches of other countries to implementing a trauma system.

**Keywords:** trauma care; trauma centre; prehospital care; emergency services

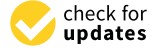



## 1. Introduction

Globally, injury is a public health issue causing physical, psychological, and functional problems for affected individuals [1,2]. Injury causes 9% of the world's mortality [3], accounting for 5.8 million deaths yearly [3]. Alongside the development of road rules legislation, vehicle safety enhancements, public behaviour change campaigns, and law enforcement, improvements in trauma care systems including trauma response and injury prevention programs have driven a reduction in the trauma burden over the last two decades [4–8]. This evolution of systems ensures a clear pathway of care for patients from the point of injury to the final phase of rehabilitation through centralising trauma resources and standardised treatment approaches (Figure 1). The fundamental goals of

the system are to reduce traumatic injury-related mortality and improve patients' health-related quality of life (HRQoL). A trauma system consists of different essential components in the prevention of and response to an injury event, including the prehospital response, in-hospital care, rehabilitation care, and injury prevention [9]. Different phases of care, reflecting the structure and processes of trauma care systems, are summarised in Table 1 [10].

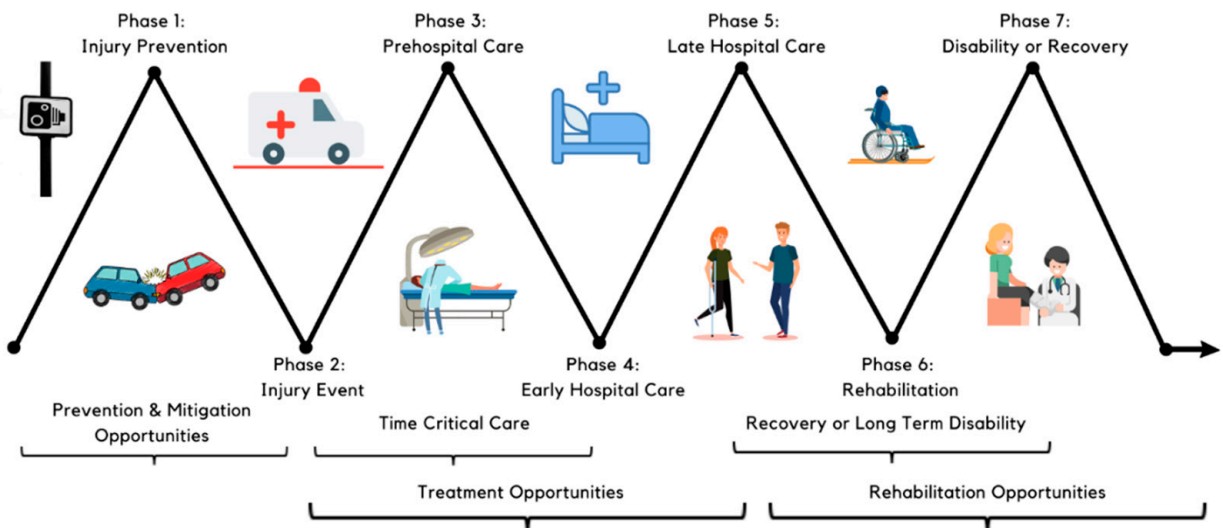

**Figure 1.** Summary of a trauma patient's journey from injury to disability or recovery if death does not occur.

**Table 1.** Shows the main phases of trauma care.

| Stage of Care | Element | Process |
| --- | --- | --- |
| Prevention | Analysing patients' data to establish the most valuable programs to prevent injury. | Road safety legislation such as alcohol screening, speed limit, and seatbelt, as well as good enforcement. |
| Prehospital | Dispatch and bystander's instructions, EMS care, triage, and transportation. | EMS Trauma protocol, fast and appropriate transportation to trauma care facility. |
| Hospital | Stabilisation and preparation for transfer to the higher-level trauma centre if needed. Definitive care (Trauma Centre or equivalent). | Activation of trauma team and preparation to receive patient. |
| Post-hospital | Rehabilitation services and home follow-up care. | Rehabilitation protocol and return to optimal activities. |

Note: EMS = Emergency Medical Service. Adapted from "Indicators of the quality of trauma care and the performance of trauma systems" by Gruen, R., Gabbe, B., Stelfox, H., and Cameron, P., 2012, British Journal of Surgery, 99(1)v, pg 99 [10].

Trauma system improvements have resulted in decreasing mortality of traumatic injury patients in North America [6,11,12], Europe [13], Asia [14,15], and Oceania [11,16]. Summarising the diverse characteristics of trauma systems being developed and implemented in different international contexts may support low- and middle-income countries (LMIC), where system development is still emerging. According to Callese et al. [17], there is no single prehospital and hospital trauma system appropriate for all LMICs. Therefore, highlighting the diversity of systemic responses to injury would provide an opportunity for LMICs to consider an approach to trauma infrastructure that is most appropriate for their resources, unique social, cultural, and geographic characteristics, and existing health care and traffic systems. By collating this information into a single review, comparison of similarities and differences is more readily apparent and highlighted.

There is limited literature describing the common and different characteristics of prehospital and hospital trauma systems in different regions. This literature review focused

on reviewing the characteristics of prehospital and hospital trauma services in different regions around the world based on the available refereed literature. Specifically, the North American, Latin American, European, Asian, and Middle East, Oceania, and African trauma systems were considered. The aim of this review was to demonstrate the variation in trauma care services that exist worldwide by showing examples from countries in all regions.

## 2. Materials and Methods

This study reviewed published literature that describes the characteristics of current prehospital and hospital trauma services from a number of international perspectives. In this literature review, articles were retrieved from PubMed/MEDLINE and EMBASE via Ovid. These two databases were chosen as their respective indexing (MeSH and Emtree) allows for precise retrieval. Reference lists were searched to identify additional studies. This review included peer reviewed articles that were published between 2000 and 20 June 2020, written in English, and contained information about the characteristics of the trauma system. Truncation and quotation marks were used to include American and British spelling and plural nouns. Different key words were used in the search to find relevant articles (See Appendix A). The search strategy was developed by the primary author in consultation with a university-based librarian and co-authors.

All articles resulted from the selected databases were uploaded to Covidence systematic review software (Covidence, Melbourne, Australia) for abstract/title screening, and full text reading after duplicated studies were removed by the Covidence software (Figure 2). The primary author (RA) reviewed the remaining studies for inclusion in consultation with the co-authors (VL and CM) in cases of uncertainty about the inclusion or exclusion of specific articles. Articles were included if they described the characteristics of the trauma system including the nature of prehospital, hospital care provided, or other components of the trauma network such as quality of trauma care and trauma registry. Grant and Booth [18] indicated that a quality assessment is not required for a literature review of this type, unlike a systematic review; however, the result of each study is presented with general commentary regarding the level of evidence. A qualitative synthesis of the literature was undertaken, comparing and contrasting the characteristics of the trauma systems described and expected.

Based on preliminary results, it was clear that there was not always sufficient information in the refereed literature to develop clear descriptions of the trauma care systems in the study region, particularly around prehospital care. Therefore, we also sought further information from grey literature, specifically annual performance reports for prehospital service providers. We selected 1–2 examples from each region to add further detail to the descriptions based on the refereed literature but did not undertake a comprehensive search of grey literature for all available reports.

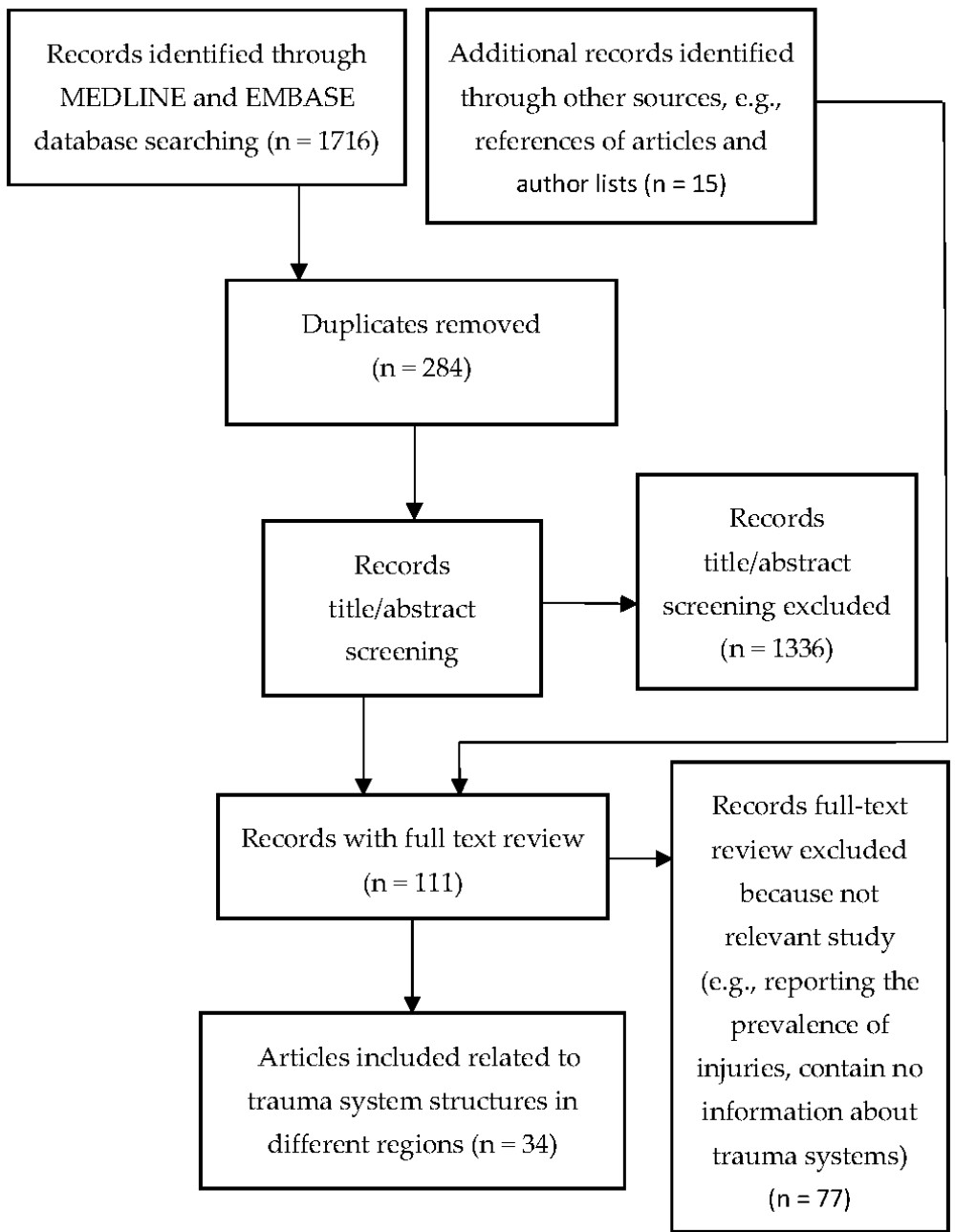

**Figure 2.** PRISMA flow diagram showing the searching and screening processes.

### 3. Results

After the abstract, title screening, and full text reading processes were completed, 34 studies that had information about the characteristics of the trauma care system in which the research was conducted were included. Articles described systems in seven regions: the North America, Latin America, Europe, Asia and Middle East, Oceania, and Africa [19–52]. While all articles provided some information about the trauma system, descriptions were not comprehensive and there was little consistency. Therefore, eight prehospital emergency medical service providers' annual performance reports were reviewed to support more detailed and comparable descriptions to be presented [53–60]. Detailed examples are from Oregon and Northern Ohio, US; British Columbia and Quebec Provinces, Canada; London, United Kingdom (UK); Emilia-Romagna, Italy; Hong Kong, China; Victoria, Australia; and Western Cape Province, South Africa.

*3.1. U.S. Trauma System*

In the early 1970s, American College of Surgeons Committee on Trauma (ACS COT) began development of the U.S. trauma system. In the late 1980s, the U.S. National Trauma Data Bank began to receive data from over 400 U.S. hospitals [19]. The U.S. has multiple independent trauma systems within its states. However, in general, the public emergency medical service system was either incorporated into fire departments or developed as a third-service separate and independent from police and fire services [20]. The U.S. prehospital rescue system is provided by a non-physician in most states [19]. Basic life support services are provided by Emergency Medical Technicians (EMT) with a limited scope of practice [20] and advanced life support clinicians/providers who are certified as EMT-intermediate or EMT-Paramedic level. EMT-paramedics are the most advanced prehospital care provider with skills and knowledge to treat the majority of prehospital conditions such as advanced airway compromise and tension pneumothorax [20]. Further, a medical director (physician) is available within the system providing off-line (indirect) communication such as protocols/practice guidelines and trauma triage guidelines, as well as online (direct) support such as direct consultation by radio or telephone communications regarding patient care.

The U.S. system is designed to transport trauma patients to a well-equipped health care facility to provide care to the injured patient [20]. The ACS COT have developed guidelines for trauma centres in the U.S., which include criteria for activation of the trauma team, team membership, equipment, and other resources [20]. The six minimum criteria for full trauma team activation as stated by the ACS COT are shown in Table 2 [61].

**Table 2.** Minimum Criteria for Full Trauma Team Activation.

- Confirmed blood pressure less than 90 mm Hg at any time in adults and age-specific hypotension in children;

- Gunshot wounds to the neck, chest or abdomen, or extremities proximal to the elbow/knee;
- Glasgow Coma Scale score less than 9 with mechanism attributed to trauma;
- Transfer patients from other hospitals receiving blood to maintain vital signs;
- Intubated patients transferred from the scene, - OR -
- Patients who have respiratory compromise or are in need of an emergent airway

  ⇒ Includes intubated patients who are transferred from another facility with ongoing respiratory compromise (does not include patients intubated at another facility who are now stable from a respiratory standpoint)

- Emergency physician's discretion

Reprinted from Rotondo, M. F., Cribari, C., Smith, R. S., and American College of Surgeons Committee on Trauma (2014). Resources for optimal care of the injured patient. Chicago: American College of Surgeons, 6. Pg 38 [61].

U.S. health care facilities usually follow the ACS COT model of hospital care for trauma patients [61]. The ACS COT defines a trauma centre as "a network of definitive care facilities that provides a spectrum of care for all injured patients" (Page 2) [61]. In the U.S., a trauma surgeon or emergency medicine physician is usually the team captain, surrounded by other team members which include trauma residents and other allied staff such as nurses, radiology, laboratory, and respiratory technicians [20].

There are several different types of trauma centres in the U.S. A Level I trauma centre as described by the ACS COT is a regional tertiary care facility with the ability to provide total care for every trauma patient from prevention to rehabilitation. A Level I trauma centre provides the highest level of surgical care to trauma patients. It must be equipped 24 h a day with the necessary resources and personnel including general, neurological and orthopaedic surgery, emergency medicine consultants, anaesthetists, and an intensive care unit (Table 3).

**Table 3.** The ACS COT trauma centre classification.

| | |
|---|---|
| Level I | Lead hospital and tertiary care centre central to the system.<br>Leads in all aspects of trauma care, from prevention to rehabilitation.<br>Must admit at least 1200 trauma patients per year or have 240 patients with an Injury Severity Score (ISS) of greater than 15 or an average of 35 patients with an ISS of more than 15 for all general surgeons taking trauma calls.<br>Either an attending surgeon or a resident at the postgraduate year 4 or 5 must be in-house 24 h a day.<br>Resident may begin resuscitation but may not substitute for the surgeon.<br>Expected that the attending surgeon will be in the emergency department within 15 min of patient arrival.<br>Hospital must document the presence of the attending surgeon at least 80% of the time.<br>While on call, surgeon must be dedicated only to that centre and can have no responsibilities at another centre.<br>Backup call schedule must be available. |
| Level II | Must be 24-h in-house availability of the attending surgeon.<br>Resident at the postgraduate 4 or 5 year or an attending emergency physician who is part of the trauma team may begin the resuscitation, but cannot substitute for the surgeon.<br>Expected that the attending surgeon will be in the emergency department within 15 min of patient arrival.<br>Hospital must document the presence of the attending surgeon at least 80% of the time.<br>While on call, the surgeon must be dedicated only to that centre and can have no responsibilities at another centre.<br>Backup call schedule must be available. |
| Level III | On-call surgeon must be available in the emergency department within 30 min of patient arrival.<br>Must demonstrate a commitment to injury prevention, outreach activities to the local community, and education to all providers involved in the care of the injured patient. |
| Level IV | Located in a rural setting.<br>Provides initial evaluation of injured patients.<br>24-h emergency coverage must be available by a physician. |
| Non-trauma centre | Delivers and regularly provides care to less severely injured patients<br>Exists within the trauma system. |

Reprinted from Rotondo, M. F., Cribari, C., Smith, R. S., and American College of Surgeons Committee on Trauma (2014). Resources for optimal care of the injured patient. Chicago: American College of Surgeons, 6. Pg 17–20 [61].

*3.2. Canadian Trauma System*

Similar to the U.S., Canada also has multiple independent trauma systems over different provinces. In the early 1990s, a regional trauma registry was established in Quebec province [31]. Prehospital care is mainly provided by an advanced life support paramedic, especially in large urban areas. Basic life support personnel are also available in some cities (e.g., Montreal). The prehospital service is supported by air-ambulance services including fixed-wing planes and helicopters, with a physician and minimum advanced life support paramedic [41]. The system has established bypass and triage criteria for trauma patients. The activation of the trauma team is based on criteria related to the patient's condition such as patients with a Glasgow Coma Scale score less than 8 and patients requiring a blood transfusion [30].

In Canada, Ontario province has an exclusive trauma system with no designated Level III or IV. Other provinces such as British Columbia and Quebec provinces have an all-inclusive trauma system with five different levels of trauma centres ranging from Level I to Level V. Trauma centres in Canada are accredited by the Trauma Association of Canada. Level I describes the service with the central role in the provincial trauma system providing the highest level of trauma care with all essential specialties available. Level II is similar to Level I but without research programs and some trauma training. Level III provides initial care before transferring patients to Levels I or II trauma centres. Level IV is an urban hospital, whereas Level V is a small rural community hospital [41]. In some Canadian provinces such as Quebec, Level III and Level IV trauma centres are normally used for early stabilisation of the injured patient before transfer to a higher-level trauma centre. The distribution of Level I and Level II centres is organised so that approximately 77.5% of the Canadian population live within a 1-h road trip to these centres [41].

### 3.3. Latin American Trauma System Examples

Mexico and Brazil represent one-third of the overall South American continent. In Mexico, prehospital care is provided equally by both trained paramedics and volunteers; however, the service coverage is not comprehensive across the country [46]. With respect to the hospital trauma system, there are three different levels of trauma centres in Mexico; Level 3 is equipped with all necessary resources to treat major trauma patients, Level 2 provides management and stabilisation for the majority of trauma patients, and Level 1 hospitals lack trauma resources [46]. Doctors who trained at an advanced trauma life support level are not available in all Level 1 and 2 trauma centres, especially for hospitals located in rural areas [47].

In Brazil, prehospital care provides a basic life support ambulance service for areas of low population density (an ambulance for every 100,000–150,000 inhabitants) and an advanced life support ambulance service for areas of high population density (an ambulance for every 400,000–450,000 inhabitants) [49]. In Ribeirao Preto, prehospital care is mainly provided by basic life support response teams, advanced life-support-trained staff, and a physician. In the absence of a trauma triage protocol, decisions about the transport of injured patients are frequently made by the response team in Ribeirao Preto [48]. A helicopter emergency medical service is available in Brazil. In Sao Paulo, there are 5 Level I trauma centres, where Level I centres provide care for major trauma patients and Level 2 centres care for less severe injuries [46].

### 3.4. European Trauma System Examples

In 1991, the British Government began development of a trauma system in the UK [26]. Most of the ambulances in the UK are staffed by paramedics with training in emergency assessment and resuscitation. Prehospital care is provided by air ambulances and helicopter services that provide rapid transportation to secondary care in rural areas, as well as in urban settings in the presence of extreme traffic congestion [29]. The prehospital service uses a triage protocol that is based on the patient's physiological status and mechanism of the injury. Major trauma patients will be transported by ambulance directly to a Major Trauma Centre (MTC) within 45 min, bypassing all hospitals on route [38]. The MTCs provide the highest level of care to manage all types of trauma in the UK system. The first hospitals designated as MTCs were in 2010–2011 in London. By the year 2014, there were 26 MTCs in England's trauma system providing care for both adults and children [38].

Germany, Austria, and Switzerland operate a model similar to the American model for prehospital care including basic life support and advanced life support paramedics, with emergency physician and trauma surgeon leadership for in-hospital trauma care [25]. In Germany, the TraumaNetworkDGU was established by trauma surgeons in partnership with the German Society for Trauma Surgery in 2009, even though a trauma registry was initiated much earlier in 1993 [37]. The prehospital care system is provided by different levels of emergency medical service staff including a physician [28,36] accompanied by an EMT trained in advanced life support or ambulance crew consisting of an EMT trained in advanced life support and a driver trained in basic life support [28]. The German TraumaNetworkDGU has classified hospitals into three levels of trauma centres, including supraregional, regional, and local trauma centres. The supraregional hospitals provide the highest level of trauma care inclusive of all medical disciplines. The local trauma centres are the smallest trauma care facilities and are able to provide acute stabilisation to patients such as controlling bleeding [37]. In Germany, more than 95% of severely injured patients are seen by an emergency physician on scene; 34.6% of those are brought to a trauma care hospital by helicopter within an average of 70 min [24].

In France, while there is no national trauma system or trauma registry, some regions have a local system such as the Northern French Alps Trauma network, which was implemented in 2007 with a registry of the system operational since 2009. The system has similarities to the American system although the prehospital protocol to triage patients functions according to a three-level system, where the severity of trauma patients is cat-

egorised as A, B, or C with Grade A representing the most critical patients [25]. In 2018, Gauss et al. [62] proposed a strategic plan for a national trauma system including a national trauma registry to be implemented within 10 years.

In Greece, there is currently no trauma system, trauma registry, or quality control program. Prehospital care is provided by basic life support and advanced life support EMTs (who have received a 2-year comprehensive training program), as well as physicians for special mobile units [39]. There are no MTCs to treat trauma patients [39]. In The Netherlands, prehospital care is mainly provided by ambulance personnel including an ambulance driver who has some medical training and a qualified paramedic, with strict protocols for the majority of emergency conditions [27]. In 1990, the Dutch Trauma Society recommended that all hospitals should be categorised according to their ability to provide care for trauma patients. In 1999, this contributed to the creation of 10 trauma centres across The Netherlands. Emilia-Romagna, Italy has around eighty hospitals providing adult acute care services under an organised trauma system since 2006 with no paediatric trauma centres. This system is built around three hubs, similar to Level I trauma centres and evenly spread throughout the territory. The emergency medical services are comprised of ground ambulances operated by crews with varying medical skills and three anaesthetist-manned helicopters that run only during the day [63].

### 3.5. Asian and Middle Eastern Trauma System Examples

In Hong Kong, there is only one level of trauma centre that is the equivalent to Level I centres in the U.S, with five designated trauma centres operating currently across Hong Kong. Until 2018, each of these centres had its own trauma registry, in lieu of a national trauma registry. However, a prehospital protocol that aligns with the trauma care system is now in place [45]. The prehospital protocol activates a team approach to trauma care when a trauma patient meets the criteria of systolic blood pressure < 90 mmHg, respiratory rate 10< or >29 breaths per minute, and GCS ≤ 13 [23].

In Japan, a prehospital care system was established in the early 1990s. Japan implemented a prehospital model of care unlike the U.S. and European models of care, in that they do not have standardised training for EMTs such as basic life support or advanced life support. The prehospital service is mainly provided by the fire defence system that is operated by the government. Hospital care and organisation in Japan is also different from the U.S. model of the trauma system. There are no dedicated trauma centres; patients with severe injuries are transported to either a university hospital's emergency department or lifesaving emergency centres [33].

In Saudi Arabia, the prehospital service is provided by two levels of prehospital personnel: EMTs and paramedics alongside physicians for severely injured patients in some cities [43]. Saudi Arabia has no trauma system; however, there are two MTCs in the capital city Riyadh that provide the care equivalent to a U.S. Level I trauma centre. There is no trauma system in Iran. Formal prehospital care is delivered by EMTs; however, the majority of trauma patients are transported to hospital by private vehicle (family members or friends) with no formal designated trauma centres [40].

### 3.6. Oceania Trauma System Examples

The Oceanic region includes countries such as Australia and New Zealand. Australia is comprised of states and territories with each operating its own trauma registry. This includes the New South Wales trauma system that was established in 1991, South Australian Trauma System that was established in 1997, and Victorian State Trauma System that was established in 2001. In Victoria, the development of a trauma system began with three designated MTCs and the implementation of a trauma registry. The system has features including trauma response management, triage of trauma patients and protocols for transfer, improved transfer services including retrieval, ongoing education and training, integration of rehabilitation services, ongoing technology developments, and continuous

research and quality management. The Australian prehospital care system is provided by a non-physician [19].

Similar to England's prehospital service, all trauma patients must be transported to an MTC within 45 min, bypassing all other hospitals. Activation of the trauma system in Victoria is based on criteria including the mechanism of injury such as vehicle roll over, a fatality in the same vehicle, ejection from the vehicle, motorcycle accident, or cyclist impact at 30 km or greater per hour [23]. Since Australia has large remote areas, the Royal Flying Doctor Service of Australia has been providing prehospital support for remote communities since 1928. The service also delivers primary health clinics and remote (telephone) consultations across Australia.

In New Zealand, 54% of all major injuries are caused by road-traffic crashes (blunt trauma). Auckland City Hospital was the first hospital to establish a trauma service in the 1990s [22]. In 2012, the national trauma system was established, known as the Major Trauma National Clinical Network. The main goal for the New Zealand system for the period 2012–2017 was "to establish a formal national structure, to implement a national registry and to develop consistent guidelines and policies" (Page 20) [44]. It is expected that the New Zealand trauma system will become a mature system by 2022. The main prehospital care provider in New Zealand is St John's Ambulance Service, which provides around 90% of prehospital care nationally [22]. Prehospital care is provided by different levels of qualified emergency medical service professionals include basic life support and advanced life support care such as a paramedic. General practitioners also provide prehospital care in some geographical areas [22]. Volunteer officers also constitute part of the New Zealand prehospital care particularly in rural areas [32].

### 3.7. African Trauma System Examples

The South African trauma system is more evolved than other African countries. In South Africa, prehospital care incorporates preventative strategies from the Government Healthcare Plan in relation to trauma-associated accidents, such as gun control and substance abuse. The service is provided by both government and private companies, with the majority of metropolitan areas served by helicopter ambulance and private fixed-wing air evacuation [42]. The prehospital medical staff include a mix of disciplines, including basic ambulance assistant, ambulance emergency assistant, critical care assistant, and paramedic (advanced life support practitioner). Across the country, there are seven specialised, highly equipped trauma care units that can provide specialist general surgical care in a timely manner [21]. Most aspects of the South African public hospital system are "modelled on the former UK system of being casualty departments" [42].

Unlike South Africa, other African countries such as Nigeria have less structured prehospital trauma systems with no trauma protocols or triage guidelines in place. Police and bystanders are usually the first responders [51]. In Malawi, there is no formal system of prehospital trauma care, and a lack of hospital trained staff, trauma resources, or organised hospital trauma system to treat traumatic injury patients [52]. Tables 4 and 5 summarises the characteristics of prehospital and hospital trauma care systems, respectively, across six regions based on information from the 34 articles and eight example annual reports (North America, Latin America, Europe, Asia, Oceania, Africa).

**Table 4.** Characteristics of prehospital trauma care system examples across the seven regions (from refereed literature and example Annual Reports).

| Region | US | Canada | Latin America | Europe | Asia | Oceania | Africa |
|---|---|---|---|---|---|---|---|
| Name of the sites | Oregon, U.S. | British Columbians, Canada | Sao Paulo, Brazil | London, UK | Hong Kong | Victoria state, Australia | Western Cape Province, South Africa |
| Size of Population | 4.1 million | 4.4 million | 12 million | 8.7 million | 7.4 million | 6 million | 6.2 million |
| Total area (km$^2$) | 250,000 | 944,000 | 1521 | 100,000 | 1100 | 227,000 | 129,462 |
| Service provider | Private companies | British Columbians Ambulance Service | SAMU-SP | London ambulance service | HK Emergency Ambulance Services | Ambulance Victoria | Western Cape EMS |
| Funding | Private | Government | Government | Government | Government and non-government | Government | Government and private |
| Service fee | Yes | Yes | NA | Yes | Yes | Yes | Yes |
| Number of ambulance/stations | NA | 184 stations. 500 ambulances, 62 support vehicles, 46 bikes, and 2 gators. | 77 stations | 70 stations | 368 ambulances, 4 mobile casualty treatment centres, four village ambulances and 36 ambulance-aid motorcycles | NA | 250 ambulances |
| On field EMS personnel | EMR, EMT, Advanced EMT EMT-Intermediate EMT-I Paramedics, physicians, firefighters | Physician, paramedic, nurse, emergency medical responders, primary care paramedics, advanced care paramedics | Physician, ALS, BLS | Flight paramedic, Paramedics, EMT | Physician, emergency medical assistant II, EMT-I, EMT-Paramedic | MICA paramedic/MICA/ Paramedic/ACO | BLS/ILS/ALS/ Paramedic [53] |
| Trauma protocol | Yes | Yes | NA | Yes | Yes | Yes | Yes |
| Field triage protocol | Yes | Yes | NA | Yes | Yes | Yes | Yes |
| Ambulances to treat and transport people | NA | Ambulances, cars, bikes, gators, helicopter | NA | Ambulances, cars, motorcycles, bikes, helicopter | Ambulances, motorcycles, helicopter | Ambulances, cars, motorcycles, bikes, helicopter | Ambulances, cars, bikes, helicopter |
| Mean response time | <14 min [54] | <9 min (65%) red flag incident [55,56] | 27 min for 98% of the incident [50] | 8 min (69.19%) priority 1 call [59] | 12-min (91.8%) [57] | <15 min for (85%) [58] | <15 min for (65%) urban response [53] |

Note: U.S. = United States; EMS = Emergency Medical Service; NA = Not available; EMR = Emergency Medical Responders; EMT = Emergency Medical Technicians; UK = United Kingdom; HK = Hong Kong; MICA = Flight Mobile Intensive Care; ACO = Ambulance Community Officer; BLS = Basic Life Support; ILS = Intermediate Life Support; ALS = Advanced Life Support; SAMU-SP = Mobile Emergency Care Service of Sao Paulo.

**Table 5.** Characteristics of hospital trauma care system examples across the seven regions (from refereed literature and example Annual Reports).

| Region | U.S. | Canada | Latin America | Asia | Oceania | Europe | Africa |
|---|---|---|---|---|---|---|---|
| Name of the sites | Northern Ohio | Quebec province | Sao Paulo, Brazil | Hong Kong | Victoria state, Australia | Emilia-Romagna, Italy | Western Cape, South Africa |
| Size of population | 4.5 million | 8.2 million | 12 million | 7.4 million | 6 million | 4.5 million | 6.2 million |
| Total area (km$^2$) | 22,000 | 1667 million | 1521 | 1100 | 227,000 | 22,000 | 130,000 |
| Trauma system implementation year | 2010 | 1992 | NA | 2000 | 2001 | 2006 | NA |
| Designated hospital for trauma | Yes | Yes | Yes | Yes | Yes | Yes | Yes |
| Designated hospital Level | 1 Level I, 2 Level II, and 1 Level III TC | 5 Level I TC and 25 secondary care centres (can treat severe and multi-trauma patients) | 5 Level I TC | 5 Designated TC | 3 MTCs, 9 Metropolitan Trauma Services | 3 TC equivalents to Level I TCs | TC |
| Top mechanisms of injury | MVC | Blunt injury | Blunt injury [48] | Blunt injury (83.9%) | MVC | Blunt injury | Violence [34] |
| Age group | 21–40 | <65 | 35.7 ± 20.6 | Median age 45 years | 24–45 | 17–44 | <40 |
| Mortality rate | Decreased by 2% (4 years follow trauma system implementation) [60] | Reduced by 43% (10 years follow trauma system implementation) [31,35] | NA | Decreased by 3% (5 years study post trauma system implementation) [14] | Preventable death reduced by 8% (2 years post the new trauma system implementation) [64] | Reduced by 30% in the most injured third of patients (5 years study post trauma system implementation) [63] | NA |
| Trauma registry | Yes | Yes | NA | Yes | Yes | Yes | Yes |
| Trauma education programs | Yes | Yes | Yes | NA | Yes | NA | NA |
| Quality assurance program | Yes | Yes | Yes | Yes | Yes | Yes | Yes |

Note: U.S. = United States; NA = Not Available; MVC = Motor Vehicle Crash; MTC = Major Trauma Centre; TC = Trauma Centre.

## 4. Discussion

This study sought to highlight the similarities and differences in prehospital and hospital trauma systems around the world by a qualitative synthesis of information available in refereed literature from 2000 to 2020 supplemented with information from examples from countries in all regions. The burden of traumatic injury is a major public health problem globally [65]. Prevention and mitigation of injuries are essential components in addressing trauma. Effective injury prevention has been seen through public education and awareness raising campaigns and programs implemented by government and other agencies, and in application of legislation and policies such as improving road design, road rules and regulations such as mandatory seatbelt use, speed controls, and road traffic fines [39,66–72]. Mitigation of injury is primarily through the development of robust trauma systems that include effective prehospital care and designated trauma centres.

Prehospital care provided at the scene of the trauma until the patient arrives at hospital is a clinically critical time for injured patients, with studies demonstrating that the majority of road traffic trauma deaths occur at the site of the crash, during transportation to a health facility, or within the first hour following the crash [73,74]. Studies in high-income countries such as the U.S and Europe have shown that the key factors influencing positive outcomes of road trauma patients are early intervention at the scene, with effective resuscitation and transporting victims to an appropriate health facility based on the patient's need [75,76]. Evidence shows that the presence of a trauma system is associated with decreased prehospital time [77], effective patient triage [78] and increased prehospital notification [31]. Furthermore, when there was no delay in transportation to a hospital, mortality was found to be reduced [79]. Prehospital blood transfusion is another essential factor in the care for patients experiencing massive hemorrhage. One study demonstrated that prehospital blood transfusion decreased battlefield mortality in the military [80].

In-hospital trauma care begins when a hospital receives notification from the prehospital care provider to prepare for a patient's arrival. The early activation and preparation of the trauma team prior to arrival of the patient has been seen to contribute to shorter resuscitation times and time to start emergency operations [81].

Another component of the trauma system is rehabilitation. The patient rehabilitation journey usually begins following stabilisation of the patient's injury and is based on the patient's needs. The fundamental aim of trauma care rehabilitation is to return an injured patient to their preinjury health status. The rehabilitation team normally assesses patients to make a plan and set goals that maximise the patient's benefits from rehabilitation care. The rehabilitation of injured patients could include a number of specific interventions such as neuropsychological assessment, physical therapy, nutritional evaluation, pain control, psychological support, and occupational therapy [61]. The level of rehabilitation required by each injured patient differs; for example, patients with brain injury usually require many levels of rehabilitation. The length of rehabilitation is also determined by factors including the number and type of injuries, and the patient's age, preinjury health status, and overall health [61].

Despite the burden of traumatic injuries affecting LMICs, there remains much scope to develop mature trauma systems in those countries. Most of the refereed literature reported on developed countries with similar prehospital services, for example, prehospital trauma protocols and field triage protocols were present in all the included examples. Trauma registries and quality assurance programs were present in most regions described in the literature; however, among high-income countries, the designated hospital levels differed from one region to another. For example, in the U.S. and Germany, trauma care facilities consisted of five and three different levels of trauma centres ranging from Level I to Level IV and supraregional to local trauma centres, respectively. African countries such as Nigeria and Malawi have no designated trauma centres. Japan does not designate hospitals as trauma centres, but directs all trauma patients to university hospitals.

The international standard of emergency medical services average response time in urban areas is 8 min or less for red flag incidents such as traumatic injury and cardiac arrest

cases [82]. Our study observed diversity of average response time targets. For example, the average response time in British Columbia, Canada and Victoria, Australia were <9 min (in 65% of cases) and <15 min (in 85% of cases) for lights and sirens incidents, respectively. What was not clear from the literature was how the different factors could influence longer response times of emergency medical services in different countries [83]. However, the literature supports the conclusion that helicopter emergency medical services is important in providing rapid transportation to major trauma centre [84].

The fundamental goal of a mature trauma care system is to match the needs of the injured to the most appropriate level of care in a geographic region. This review has contributed to the literature by describing the characteristics of prehospital and hospital trauma systems in different regions in a single review so that commonalities and differences are more readily apparent and highlighted. Understanding the differences in trauma systems and service approaches used by other countries can help LMICs consider improvements to their trauma system structure to align with their own domestic characteristics and available resources. Adopting a systems approach has contributed to reducing trauma-related mortality in developed countries [85]. Further research could examine the effectiveness of different system structures in reducing mortality and morbidity across these regions.

This study had several limitations. First, the literature did not provide a comprehensive account of trauma systems in all countries and regions globally, nor did it provide sufficient detail about prehospital care in most cases. Therefore, our study represents a selective review of trauma services in different regions around the world. Second, searching for grey literature for selected examples of prehospital care could not be inclusive of all countries with the resources available. Furthermore, the bias towards availability of information from developed countries rather than LMICs was observed in the grey literature too. Third, the information present was current at the time of the review but given the evolving nature of trauma systems especially in LMICs may change. Fourth, we included only studies published in English; therefore, this review will have missed articles and reports published in languages other than English. Finally, quality assessment was not performed for the included studies; however, this was not required for such a literature review [18].

## 5. Conclusions

This review highlights developments in prehospital and hospital trauma services over the last two decades. Trauma systems were first developed in the U.S. in the 1970s, followed by development of European, Canadian, and Oceania systems in the early 1990s. There is variety in the characteristics of trauma care services within countries and that it is not possible to describe all trauma systems comprehensively. European countries such as France and Greece are still forming an integrated system. Additionally, LMIC countries such as Saudi Arabia and Iran are still in the formative phases with respect to their development of a trauma system. This review aims to enable countries with immature trauma systems to consider the similarities and differences in approaches of other countries to implementing a trauma system and how they could inform their own directions.

**Author Contributions:** This paper was developed by all authors. R.J.A. prepared the first draft of this research paper and revised comments from V.L. and C.M. All authors have read and agreed to the published version of the manuscript.

**Funding:** This research received no external funding.

**Institutional Review Board Statement:** Not applicable.

**Informed Consent Statement:** Not applicable.

**Data Availability Statement:** The datasets used and/or analysed during the current study are available from the corresponding author on reasonable request.

**Conflicts of Interest:** The authors declare no conflict of interest.

## Appendix A

**Table A1.** Search strategy for PubMed/MEDLINE and EMBASE via Ovid (search conducted: 20 June 2020).

| Search | Query | Records Retrieved |
|--------|-------|-------------------|
| **#1** | ('major trauma*' or 'major injur*' or 'trauma patient*' or 'injury patient*' or 'injured patient*' or 'traumatic injury*' or 'multiple trauma' or 'multiple trauma injur' or 'serious injur*').mp. [mp=title, abstract, original title, name of substance word, subject heading word, floating sub-heading word, keyword heading word, organism supplementary concept word, protocol supplementary concept word, rare disease supplementary concept word, unique identifier, synonyms] | 56,752 |
| **#2** | "Wounds and Injuries"/ | 76,736 |
| **#3** | ('trauma system*' or 'major trauma cent*' or 'non-trauma cent*' or 'trauma registr*' or 'trauma care*' or 'trauma service*' or 'metro trauma service*' or 'regional trauma service*' or 'rural trauma service*' or 'trauma prevention*' or 'services for trauma' or 'trauma evaluation*' or 'evolution of trauma' or 'implementation of trauma*' or 'trauma implementation*' or 'implementation of trauma').mp. [mp=title, abstract, original title, name of substance word, subject heading word, floating sub-heading word, keyword heading word, organism supplementary concept word, protocol supplementary concept word, rare disease supplementary concept word, unique identifier, synonyms] | 10,723 |
| **#4** | Trauma Centers/ | 10,571 |
| **#5** | 1 and 2 and 3 and 4 | 942 |
| **#6** | limit 5 to (english language and yr="2000 − Current") | 776 |

**Table A2.** Search strategy for EMBASE 1947-present (Ovid). Search conducted: 20 June 2020.

| Search | Query | Records Retrieved |
|---|---|---|
| **#1** | ('major trauma*' or 'major injur*' or 'trauma patient*' or 'injury patient*' or 'injured patient*' or 'traumatic injury*' or 'multiple trauma' or 'multiple trauma injur' or 'serious injur*').mp. [mp=title, abstract, heading word, drug trade name, original title, device manufacturer, drug manufacturer, device trade name, keyword, floating subheading word, candidate term word] | 74,435 |
| **#2** | injury/ | 386,523 |
| **#3** | ('trauma system*' or 'major trauma cent*' or 'non-trauma cent*' or 'trauma registr*' or 'trauma care*' or 'trauma service*' or 'metro trauma service*' or 'regional trauma service*' or 'rural trauma service*' or 'trauma prevention*' or 'services for trauma' or 'trauma evaluation*' or 'evolution of trauma' or 'implementation of trauma*' or 'trauma implementation*' or 'implementation of trauma').mp. [mp=title, abstract, heading word, drug trade name, original title, device manufacturer, drug manufacturer, device trade name, keyword, floating subheading word, candidate term word] | 13,325 |
| **#4** | emergency health service/ | 99,193 |
| **#5** | 1 and 2 and 3 and 4 | 1071 |
| **#6** | limit 5 to (english language and yr="2000 − Current") | 940 |

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
