# Peer review of "International Perspectives of Prehospital and Hospital Trauma Services: A Literature Review"

_traumacare, doi:10.3390/traumacare2030037_

Round 1

Reviewer 1 Report

The article is interesting, and well written.

I suggest modifying all the tables, to make them easier to read, right now, it is difficult to manage these large tables.

Author Response

Reviewer comments: The article is interesting, and well written. I suggest modifying all the tables, to make them easier to read, right now, it is difficult to manage these large tables.

Authors response: Thank you for reviewing our article. We do note there is a lot of information in the tables. We are happy to work with the journal to ensure the final formatting of the tables makes them as easy to read as possible.

Reviewer 2 Report

Dear authors, please check the following considerations:

1. line 54 check the reference n 12 if it is really referred to the european experience,.

2. line 159: please insert the meaning of ACS COT, the first time you mentioned it;

3. line 245: in the results you mentioned the experience of Emilia Romagna and Lombardy; unfortunately, in the paragraph 3.4 you didn't perform any refernce to the italian experience;

4. line 54: the reference 13 refers to the Lombardy experience, but there is no mention to Lombardy in table V, nor in further paragraph.

Author Response

Reviewer comments: Dear authors, please check the following considerations: 1. line 54 check the reference n 12 if it is really referred to the european experience,.

Authors’ response: We have corrected this reference. It refers to the US experience.

Reviewer comments: 2. line 159: please insert the meaning of ACS COT, the first time you mentioned it;

Authors’ response: We have inserted the meaning of ACS COT “American College of Surgeons Committee on Trauma (ACS COT)”

Reviewer comments: 3. line 245: in the results you mentioned the experience of Emilia Romagna and Lombardy; unfortunately, in the paragraph 3.4 you didn't perform any refernce to the italian experience;

Authors’ response: The Emilia Romagna, Italy experience was mentioned in Table V - Characteristics of hospital trauma care system examples across the seven regions (from 372 refereed articles and example Annual Reports). Section “3.4. European trauma system examples” provides an example from different Europe countries, including British, Germany, France, Greece, and Netherlands. We have now added a section about Emilia Romagna, Italy.

Reviewer comments: 4. line 54: the reference 13 refers to the Lombardy experience, but there is no mention to Lombardy in table V, nor in further paragraph.

Authors’ response: Reference 13 is part of the background introduction showing epidemiological data of hospitalised severe trauma patients, rather than being included as an example in Table V or the discussion. This reference has been replaced with a new one that shows how trauma system improvements have resulted in decreasing mortality of traumatic injury patients in a European country.

Reviewer 3 Report

The authors provide a narrative review with a comfortable lecture of the manuscript.

Overall, the review article is as nice summary of International Prehospital and Hospital Trauma Services, but needs minor revisions including a more comprehensive section of trauma services in Europe.

It would be interesting to discuss the importance of pre-hospital transfusion in patients with severe trauma and the countries that have implemented this.

Another aspect would be the importance of air rescue means (HEMS) for the patient with polytrauma.

Author Response

Reviewer comments: The authors provide a narrative review with a comfortable lecture of the manuscript. Overall, the review article is as nice summary of International Prehospital and Hospital Trauma Services, but needs minor revisions including a more comprehensive section of trauma services in Europe.

Authors’ response: We have extended the European trauma services section by adding a paragraph about Emilia Romagna, Italy.

Reviewer comments: It would be interesting to discuss the importance of pre-hospital transfusion in patients with severe trauma and the countries that have implemented this.

Authors’ response: We added a discussion about the importance of pre-hospital transfusion in patients with severe trauma (page 12).

Reviewer comments: Another aspect would be the importance of air rescue means (HEMS) for the patient with polytrauma.

Authors’ response: Our review highlighted the availability of HEMS in some countries for rapid transportation of patients with critical and multi-injury such as in the UK, Canada, and South Africa (pages 7, 8, and 10). We have now added a comment in the discussion to highlight that helicopter emergency medical services is important in providing rapid transportation to major trauma centre (page 13).